# Keeping Models Consistent between Pretraining and Translation for Low-Resource Neural Machine Translation

**Wenbo Zhang** [1,2,3] 🆔, **Xiao Li** [1,2,3,*], **Yating Yang** [1,2,3], **Rui Dong** [1,2,3] **and Gongxu Luo** [1,2,3]

[1]  Xinjiang Technical Institute of Physics & Chemistry, Chinese Academy of Sciences, Urumqi 830011, China; zhangwenbo16@mails.ucas.edu.cn (W.Z.); yangyt@ms.xjb.ac.cn (Y.Y.); dongrui@ms.xjb.ac.cn (R.D.); luogongxu17@mails.ucas.edu.cn (G.L.)

[2]  University of Chinese Academy of Sciences, Beijing 100049, China

[3]  Xinjiang Laboratory of Minority Speech and Language Information Processing, Urumqi 830011, China

[*]  Correspondence: xiaoli@ms.xjb.ac.cn

**Abstract:** Recently, the pretraining of models has been successfully applied to unsupervised and semi-supervised neural machine translation. A cross-lingual language model uses a pretrained masked language model to initialize the encoder and decoder of the translation model, which greatly improves the translation quality. However, because of a mismatch in the number of layers, the pretrained model can only initialize part of the decoder's parameters. In this paper, we use a layer-wise coordination transformer and a consistent pretraining translation transformer instead of a vanilla transformer as the translation model. The former has only an encoder, and the latter has an encoder and a decoder, but the encoder and decoder have exactly the same parameters. Both models can guarantee that all parameters in the translation model can be initialized by the pretrained model. Experiments on the Chinese–English and English–German datasets show that compared with the vanilla transformer baseline, our models achieve better performance with fewer parameters when the parallel corpus is small.

**Keywords:** low-resource neural machine translation; monolingual data; pretraining; transformer

---

## 1. Introduction

Neural machine translation (NMT), which is trained in an end-to-end fashion [1–4], has become the mainstream of machine translation methods, and has even reached the human level in some fields [5–7]. However, almost all of these achievements rely on large-scale parallel corpora. When the parallel corpus is small, neural machine translation may have poor performance [8,9]. Therefore, how to achieve high-quality translation through abundant monolingual data with small-scale parallel corpus or even zero parallel corpus has attracted the attention of more and more researchers [10–13]. An NMT model generally contains an encoder and a decoder [1]. The encoder encodes source tokens into intermediate representations, and then the decoder generates the target tokens by the intermediate representations and previous target tokens. Language models [14] trained on monolingual data can give the probability of the next word by the previous words. Because of the similarity between language models and NMT models, many research studies use language models to improve low-resource NMT. Back translation [15] is another method which can improve low-resource NMT through generating additional synthetic parallel data.

In recent years, pretraining language models such as BERT [16] and GPT [17] have shown great superiority in natural language understanding tasks, especially when there are little supervised data available. Masked language modeling [16], in which a transformer encoder is trained by predicting the masked tokens in a sentence, can learn rich semantic information in sentences and has been proved to be an excellent pretraining model [16,18]. Cross-lingual language modeling (XLM) [18] is the first to apply pretraining models to low-resource and zero-resource neural machine translation. For low-resource semi-supervised neural machine translation, XLM first trains a transformer encoder on both source and target language monolingual data through masked language modeling, and then a pretrained model is used to initialize the encoder and decoder of transformer. The knowledge in abundant monolingual data is transferred to NMT model. Therefore, the initialized transformer only needs a small amount of parallel corpus data to fine tune and will achieve satisfactory performance after fine-tuning. However, there is a small flaw that the decoder of transformer has more parameters than the encoder because the decoder has additional layers, so not all parameters in the decoder can be initialized by the pretrained model. The mismatch between the pretraining model and NMT model will lead to the degradation of the influence of the pretrained model. MASS [19] proposed a new pretraining task to replace the masked language modeling. This new pretraining task is based on the whole transformer, so it achieved better performance. In this paper, we still use the mask language modeling as the pretraining task, but we use two transformer variants instead of the vanilla transformer as the translation model. One of these translation models is layer-wise coordination transformer [20] and the other is called consistent pretraining translation transformer. Both models are able to ensure that all parameters can be initialized by the pretrained model.

We evaluate our models on Chinese–English and English–German translation tasks. Experimental results show that: after being initialized by the same pretrained model, our models perform better when the parallel corpus is small (less than 1 million). Precisely, our contributions are as follows:

1. In order to keep models consistent between pretraining and translation, we propose to use the layer-wise coordination transformer to replace the vanilla transformer as the translation model.
2. Based on the vanilla transformer and the layer-wise coordination transformer, we propose a consistent pretraining translation transformer, which obtains better performance in the pretraining fine-tuning mode.
3. Experimental results show that our models can get better performance wtih fewer parameters under low-resource conditions.
4. Which is more important, the source language monolingual data or the target language monolingual data? is cross-language pretraining necessary? We use ablation experiments to further study these problems.

Section 2 shows the related works; Section 3 introduces the background knowledge about the pretraining fine-tuning mode in translation tasks; Section 4 presents the details of the layer-wise coordination transformer and consistent pretraining translation transformer; Section 5 describes the details of our experiments; finally, the conclusion is drawn in Section 6.

## 2. Related Works

As part of statistical machine translation (SMT) [21], language models can make use of target language monolingual corpora to assist the generation of target language in the process of translation, so it is naturally applied to NMT. Gulcehre et al. [22] integrate a language model into RNN-based NMT model through shallow and deep fusion. Similarly, Skorokhodov et al. [23] use gated fusion to combine a language model and a transformer-based NMT model [4]. Xia et al. [24] use two dual translation models as an autoencoder language model in which a translation model is the encoder and the other reverse translation model is the decoder to exploit monolingual corpus. A large number of research studies [22–26]

show that the language models have a great positive impact on neural machine translation. However, most of them need to modify the architecture of NMT to combine the translation models and the language models. As a result, the whole neural machine translation system becomes very complex, which is inconsistent with the original intention of neural machine translation, an end-to-end system.

Back translation [17], a data augmentation method in the field of machine translation, using a reverse translation model and target language monolingual corpus to produce synthetic parallel corpus, has proven to be an effective method [15,27]. Most semi-supervised methods in NMT, especially back translation, only exploit target-side monolingual data. Zhang et al. [28] use a self-learning algorithm and a multi-task learning framework to exploit source-side monolingual data. Both Cheng et al. [11] and Xia et al. [24] exploit both sides of monolingual data through dual learning and back translation. However, these models are too complex to train. Back translation is a simple but effective method. However, it also has some limitations. Although some studies have shown that the quality of synthetic parallel corpora has little influence on the translation results, it is difficult to generate acceptable synthetic parallel corpora when the number of parallel corpora is too small, and a high proportion of synthetic data tends to bring too much noise [29,30].

## 3. Background

For semi-supervised neural machine translation, XLM [18] first trains a transformer encoder through masked language modeling, then initializes the encoder and decoder of the transformer with the pretrained model respectively. The initialized transformer can be trained on parallel corpus to learn the translation knowledge. The architecture of pretraining model and transformer are shown in Figure 1. We can learn that the architecture of pretraining model is same as the transformer encoder (left of Figure 1b), but the decoder (right of Figure 1b) has an additional layer, the encoder-decoder attention layer. The details of the masked language modeling and transformer-based NMT are shown in the following sections.

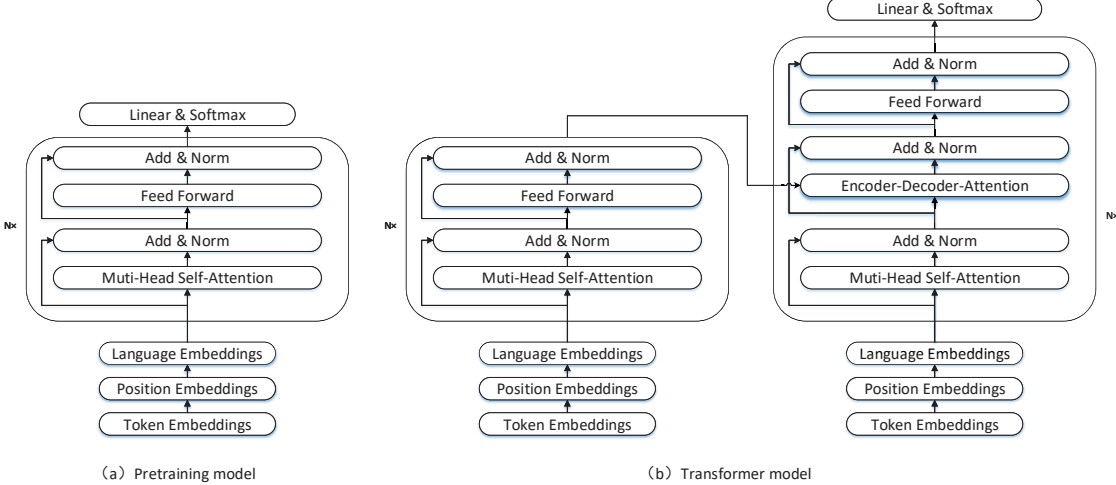

**Figure 1.** The architecture of models: (**a**) Pretraining model (**b**) Transformer model.

### 3.1. Masked Language Modeling (MLM)

The masked language modeling in XLM is similar to that in BERT [16]. Both of them take a noisy sentence as input and predict the original word in the corresponding position. The operations of adding noise are as follows: randomly extracting 15% of the tokens in the sentence; replacing them with [MASK] 80% of the time, replacing them with random tokens 10% of the time, keeping them unchanged 10% of the

time. The parameters of pretraining models are learned by predicting the masked tokens. XLM trains a transformer encoder on both the source language monolingual data and target language monolingual data to obtain a cross-lingual masked language model. In order to distinguish the sentence from the source language or the target language, XLM adds an extra language embedding layer to the input of the model.

*3.2. Transformer-Based NMT*

Transformer [4] has an encoder and a decoder. The input of encoder is the source sentences, the output of encoder is the context matrix of the source language. The decoder takes the target tokens and the context matrix of the source language as input, and gives the probability of next word in target language. Both encoder and decoder are composed of multiple identical layers. For the encoder, every layer has a self-attention sublayer and a position-wise feed-forward sublayer. For the decoder, every layer has a self-attention sublayer, an encoder-decoder attention sublayer, and a position-wise feed-forward sublayer. The self-attention sublayer and encoder-decoder attention sublayer have the same attention mechanism. The formula of the attention mechanism is as follows:

$$Attention\,(Q, K, V) = softmax\left(\frac{QK^T}{\sqrt{d_{model}}}\right) V \tag{1}$$

where $d_{model}$ is the dimension of hidden representations. The difference between the self-attention sublayer and encoder-decoder attention sublayer is that the parameters for calculating attention mechanism are different. In the self-attention sublayer of the encoder, $Q = K = V = [x_1; \ldots; x_n]$, $x$ is a token vector in the source sentence, $n$ is the length of source sentence. In the self-attention sublayer of the decoder, $Q = [y_1; \ldots; y_m]$, $y$ is a token vector in the target sentence, $m$ is the length of target sentence. For the j-th token in the target sentence, $K = V = [y_1; \ldots; y_j]$. In the encoder-decoder attention sublayer of the decoder, $Q = [y_1; \ldots; y_m]$, $K = V = [x_1; \ldots; x_n]$, where $x$ is the token vector by the last layer output of the encoder. Please refer to Vaswani et al. [4] for more details.

The optimization object of transformer-based NMT is as follows:

$$\arg\max_{\theta} \sum_{t=1}^{m} logP\,(y_t = k | x, y_{<t}, \theta) \tag{2}$$

where $\theta$ is the parameter of the transformer, $k$ is the t-th token in the target sentence.

## 4. Our Models

Following XLM [18], we use the masked language modeling as the pretraining object. From Figure 1, we can see that the encoder of transformer can be perfectly initialized by the pretrained model, but the pretrained model can only initialize the first and third sublayers of decoder in transformer. The encoder-decoder attention sublayer in the transformer decoder can not be initialized by the pretrained model. So, in our works, we consider layer-wise coordination transformer and consistent pretraining translation transformer as the NMT model. The layer-wise coordination transformer only has an encoder. Its architecture is exactly the same as the pretrained model. The consistent pretraining translation transformer has both encoder and decoder, and the encoder and decoder have the same network architecture. Therefore, both our NMT models can be fully initialized by the pretrained model. The overall architecture of theses two models is shown in Figure 2.

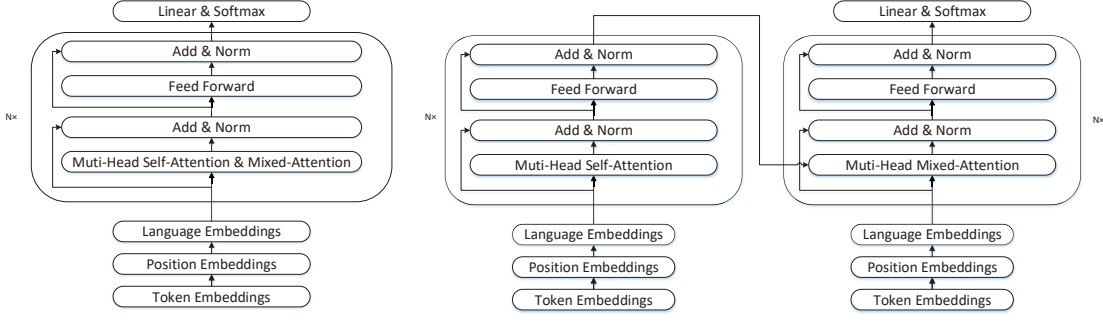

(a) Layer-wise coordination Transformer        (b) Consistent pretraining translation Transformer

**Figure 2.** The architecture of our models: (**a**) layer-wise coordination Transformer (**b**) consistent pretraining translation Transformer.

## 4.1. Layer-Wise Coordination Transformer

He et al. [20] proposed a transformer variant called layer-wise coordination transformer (LWCT) to exploit the information from low level to high level. In their model, the decoder is removed, so the encoder is not only responsible for the expression of the source language, but also for the generation of the target language. For the source language, there is no difference between this model and transformer model, both of which have the same attention mechanism. For the target language, this model uses mixed-attention to replace the self-attention and encoder-decoder attention. The mixed attention means that the target tokens can see not only previous target language tokens, but also the whole source language tokens, and it is also formulated as (1). The difference is that where $Q = [y_1; \ldots; y_m]$, for the j-th token in the target sentence, $K = V = [x_1; \ldots; x_n; y_1; \ldots; y_j]$. An illustration is shown in Figure 3. In this paper, blue represents the source language and green represents the target language. It should be noted that the decoder in transformer uses the last layer output of the encoder to generate the next token in target language, but the layer-wise coordination transformer uses every layer output of source language to obtain the next token in the target language.

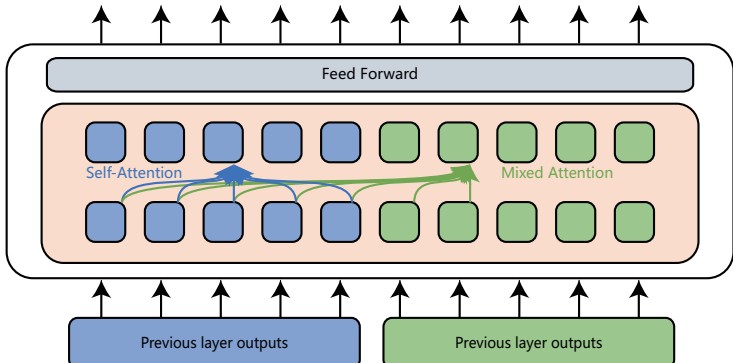

**Figure 3.** One layer in layer-wise coordination Transformer. Encoder and decoder share the same parameters. Source tokens use self-attention mechanism while target tokens use mixed-attention mechanism. The mixed-attention mechanism of each layer uses the whole source language tokens of the same layer and the previous target tokens to generate the target language outputs.

Although LWCT and the encoder of the transformer have different attention mechanisms, they still have the same number and type of parameters. In addition, pretraining only requires the encoder of

the transformer. Therefore, LWCT can be used as the translation model to keep the architecture of the pre-training model and the translation model completely consistent.

### 4.2. Consistent Pretraining Translation Transformer

The number of parameters in the neural machine translation model has a great influence on the translation results. As we can see from Figure 2, when initialized by the same pretrained model, the number of parameters of layer-wise coordination transformer is only half less than that of transformer, because layer-wise coordination transformer has only encoder and no decoder. A small number of parameters may result in poor translation performance. Besides, the layer-wise coordination transformer actually shares parameters between encoder and decoder, but for non-similar languages, such as English and Chinese, sharing parameters may have some negative effects. In order to solve these problems, we propose a new transformer variant based on the vanilla transformer and layer-wise coordination transformer, which is called consistent pretraining translation transformer (CPTT).

The consistent pretraining translation transformer has an encoder-decoder architecture. The encoder in consistent pretraining translation transformer is the same as that in transformer. The decoder also uses mixed-attention to generate target language as it does in layer-wise coordination transformer. One layer in consistent pretraining translation transformer decoder is shown in Figure 4. The differences between consistent pretraining translation transformer and layer-wise coordination transformer are consistent pretraining translation transformer does not share parameters between encoder and decoder, and uses the last layer output of the encoder to generate the target language.

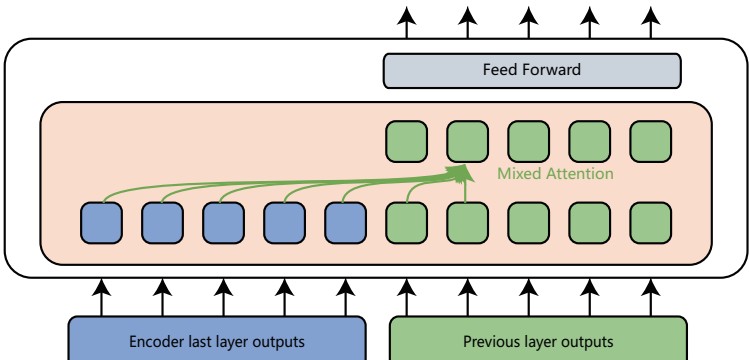

**Figure 4.** One layer in consistent pretraining translation Transformer decoder. The mixed-attention mechanism of all layers uses the source language outputs of the encoder last layer and previous target tokens to obtain the target language outputs.

### 4.3. Other Model Details

Pretraining model in XLM has a language embedding layer to distinguish source language and target language. So, in order to maintain consistency between pretraining models and translation models, all NMT models in this paper have a language embedding layer. The pretrained model shares token embedding between source language and target language, but the NMT model transformer in XLM does not share token embedding between encoder and decoder. Our consistent pretraining translation transformer maintains this setting. It means that the tokens shared by the source language and the target language may have different vector representations in the encoder and decoder.

## 5. Experiments

### 5.1. Datasets and Preprocessing

We evaluate our models on WMT17 Chinese–English and English–German datasets. English–German and Chinese–English translation tasks are high-resource cases. The parallel corpus of Chinese–English and English–German on WMT17 is shown in Table 1. To simulate the low-resource situation, we only use the News Commentary v12 parallel corpus as our training set in the translation stage. The specific data details are as follows:

**Table 1.** The parallel corpus of Chinese–English and English–German on WMT17.

| | Chinese–English | English–German |
|---|---|---|
| Parallel data | News Commentary v12 CWMT Corpus UN Parallel Corpus V1.0 | News Commentary v12 Common Crawl corpus Europarl v7 Rapid corpus of EU press releases |

**English–German** We extract the monolingual data from WMT17 monolingual corpus News Vrawl 2016, and use the first 4.5 million English and German parts of the datasets as the training data in the pretraining stage. At the same time, the News Commentary v12 English–German dataset is used as the parallel corpus, which contains 270,769 parallel sentence pairs. We use newstest2016 and newstest2017 as validation and test sets.

**Chinese–English** We use the Chinese part of all parallel corpus in CWMT corpus as the Chinese monolingual corpus, which contains 9 million sentences in total. The whole monolingual corpus News Crawl 2016 is used as the English monolingual corpus, which contains 20.6 million sentences. We only use News Commentary v12 Chinese–English dataset as the parallel corpus for Chinese–English translation, which contains 227,330 parallel sentence pairs. We use newsdev2017 and newstest2017 as validation and test sets.

All training data used in our experiments are shown in Table 2. We use Moses script (https://github.com/moses-smt/mosesdecoder) to process English and German data, and use Jieba word segmentation tool (https://github.com/fxsjy/jieba) for Chinese word segmentation. We use BPE [31] with 60,000 merge operations in the same way as XLM [18].

**Table 2.** All training data in our experiments.

| | Chinese–English | | |
|---|---|---|---|
| | Parallel data | Chinese monolingual data | English monolingual data |
| size | 227 K | 9 M | 20.6 M |
| from | News Commentary v12 | CWMT corpus | News Crawl 2016 |
| | **English–German** | | |
| | Parallel data | English monolingual data | German monolingual data |
| size | 270 K | 4.5 M | 4.5 M |
| from | News Commentary v12 | News Crawl 2016 | News Crawl 2016 |

### 5.2. Model Configurations

We use XLM to obtain the transformer baseline, and implement layer-wise coordination transformer and consistent pretraining translation transformer based on the codebase of XLM (https://github.com/

facebookresearch/XLM). For simplicity, we use LWCT to represent layer-wise coordination transformer and CPTT to represent consistent pretraining translation transformer in the rest of this paper.

We set embedding size to 512 and feed-forward hidden size to 2048 for all models. For transformer model and CPTT model, we use 6 layers for both encoder and decoder. For LWCT model, we tried two configurations, one using 6 layers encoder to ensure that the same pretrained model is used as other models, and the other using 12 layers encoder to guarantee the similar number of parameters in NMT model. During inference, we use beam search with a beam size of 4 and a length penalty of 0.6 for all NMT models.

We use Adam [32] optimizer with learning rate $10^{-4}$ for both pretraining and translation stage, and train all models on 2 V100 GPUs. For pretraining models, the batch size is set to 192 and a sequence in a batch contains 256 tokens. For NMT models, a batch contains about 20,000 tokens. Dropout [33] is set to 0.1 for all tasks. we follow XLM to set other settings.

*5.3. Results and Analysis*

As with most machine translation studies, we use BLEU [34] scores as the indicator of translation results. The translation results of different NMT models initialized by the pretrained models are shown in Table 3. According to the first three lines in Table 3, We can see that after being initialized by the same pretrained model, our CPTT performs better than XLM baseline on both zh-en and en-de translation tasks, and the LWCT with 6 layers obtains the performance close to transformer with half of the parameters. The LWCT with 12 layers even outperforms the XLM baseline by 1.46 BLEU points on en-de translation task. However, the LWCT with 12 layers requires a bigger pretrained model. We can also see clearly that our models has less parameters than transformer. Compared with the 12 layers of LWCT models, all encoder-decoder NMT models have more parameters in total, because we do not share token embedding between encoder and decoder. To compare each NMT model in detail, Figure 5 shows the results of the first 10,000 steps of the validation set on the zh-en translation task for all NMT models.

**Table 3.** BLEU scores of different NMT models initialized by the pretrained model in the test set.

| Pretraining Model | NMT Model | Encoder Param | Decoder Param | zh-en | en-de |
|---|---|---|---|---|---|
| Transformer encoder(6-layer) | Transformer(XLM) | 58.9M | 66.8M | 13.68 | 21.59 |
| | LWCT(6-layer) | 58.9M | - | 13.57 | 21.26 |
| | CPTT | 58.9M | 58.9M | 14.15 | 22.21 |
| Transformer encoder(12-layer) | LWCT(12-layer) | 77.8M | - | 13.65 | 23.05 |

An easily overlooked problem is that the improvements may come from changes to the better model, rather than relying on keeping models consistent between pretraining and translation. So, we set up a comparison experiment in which all the NMT models were not initialized by the pretrained models. The results are shown in Table 4.

We can learn that transformer performs better than all other NMT models according to Table 4. Therefore, combined with the previous experimental results, we conclude that it is necessary to keep the model consistent between the pretraining and translation stages.

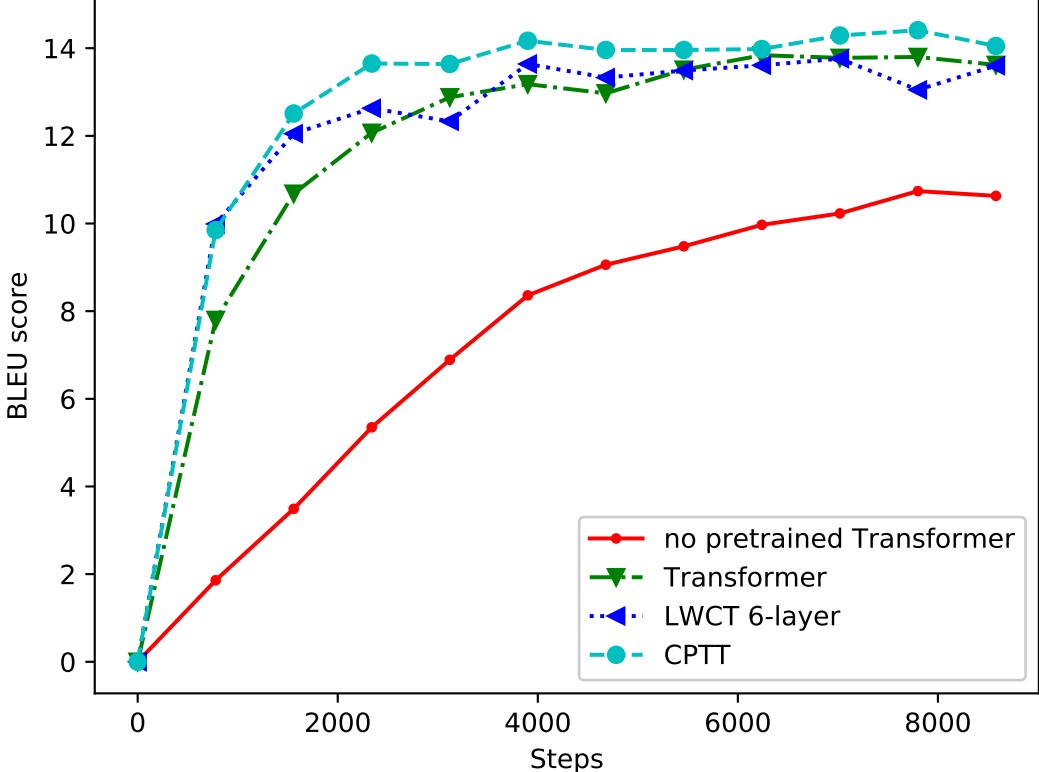

**Figure 5.** Validation results of differences NMT models on zh-en translation task.

**Table 4.** BLEU scores of different NMT models in the test set.

| NMT Model | zh-en | en-de |
|-----------|-------|-------|
| Transformer(XLM) | 10.35 | 16.84 |
| LWCT(6-layer) | 9.16 | 15.60 |
| CPTT | 10.34 | 16.45 |
| LWCT(12-layer) | 10.02 | 16.78 |

*5.4. Ablation Study*

In this section, we investigate the influence of pretrained models on encoder and decoder in NMT model, and whether cross language training is necessary in the pretraining stage. For comparison, we use a cross-lingual pretrained model to initialize only the encoder or decoder of the NMT model, only pretrained model based on source monolingual data to initialize the encoder, and pretrained model based on target monolingual data to initialize the decoder. We also tried some other initialization strategies, such as no initialization. The results of encoder-decoder NMT models are shown in Table 5, and the results of 6 layers LWCT model are shown in Table 6.

For encoder-decoder NMT models, we can obtain the following results from Table 5. Only the initializing decoder has little effect on the results of translation, but our CPTT can obtain more improvements than the transformer model. As we can see, initializing encoder is more important than initializing decoder, which means source language is more important than the target language. However, the cross-lingual model can performance better in most cases, especially using it to initialize both encoder and decoder. This situation is more obvious on en-de translation task. We think that this is because using the same pretrained model to initialize both the encoder and decoder will help NMT models

learn alignment information on parallel corpus more easily. Table 6 shows that for 6 layers of LWCT, both the pretrained model of the source language and the target language can significantly improve the translation results, the result of pretrained model based on source language is better, and the cross-lingual pretrained model has the best performance. Most of the results of LWCT are consistent with the results of encoder-decoder NMT models. So, we can conclude that in pretraining fine-tuning mode, encoder is more important than decoder; in other words, source language is more important than target language, but cross-lingual pretraining is necessary, because it can obtain a better performance.

**Table 5.** BLEU scores for encoder-decoder NMT models with different pretrained models in validation dataset. mlm_src&tgt is cross-lingual masked language model, mlm_src and mlm_tgt are monolingual masked language models.

| Encoder | Decoder | zh-en | | en-de | |
|---|---|---|---|---|---|
| | | Transformer | CPTT | Transformer | CPTT |
| - | - | 10.74 | 10.74 | 20.88 | 20.25 |
| - | mlm_tgt | 10.82 | 11.73 | 20.90 | 20.33 |
| - | mlm_src&tgt | 11.00 | 11.65 | 21.08 | 21.26 |
| mlm_src | - | 13.26 | 13.34 | 24.58 | 24.46 |
| mlm_src&tgt | - | 13.82 | 13.53 | 24.86 | 24.46 |
| mlm_src | mlm_tgt | 13.47 | 13.80 | 24.78 | 22.94 |
| mlm_src&tgt | mlm_src&tgt | 13.84 | 14.41 | 26.16 | 26.47 |

**Table 6.** BLEU scores for 6 layers LWCT model with different pretrained models in validation dataset.

| Pretrained Model | zh-en | en-de |
|---|---|---|
| - | 9.72 | 19.03 |
| mlm_src | 12.82 | 24.92 |
| mlm_tgt | 11.77 | 24.40 |
| mlm_src&tgt | 13.79 | 25.87 |

*5.5. The Influence of Parallel Corpus Size*

The method based on pretraining and fine-tuning is mainly aimed at improving the quality of low-resource neural machine translation. However, how many parallel corpora are low resource cases? When the scale of the parallel corpus becomes larger, does the model CPTT still work? In this section, we study these problems in the English-to-German translation task by changing the scale of parallel corpus. We collected all the parallel corpus data of WMT17 en-de, and then extracted different numbers of parallel corpus from them for experiments. The results are shown in Figure 6.

As we can see, all models initialized by the pretrained model achieved significant improvements, but with the increase of the scale of the parallel corpus, these improvements become smaller and smaller. When the number of parallel corpus is small, especially less than 1 million, our CPTT model can obtain more improvements than the transformer. When the scale of parallel corpus is between 1 million and 3 million, the performance of CCPT and transformer are similar. When the scale of parallel corpus reaches 4.5 million, transformer can obtain more improvements than CCPT. We think that this is because when the scale of parallel corpus becomes larger and larger, the influence of pretraining model will become smaller and smaller. Therefore, when the scale of parallel corpus is large, the transformer will obtain better performance, otherwise CCPT may be a better choice.

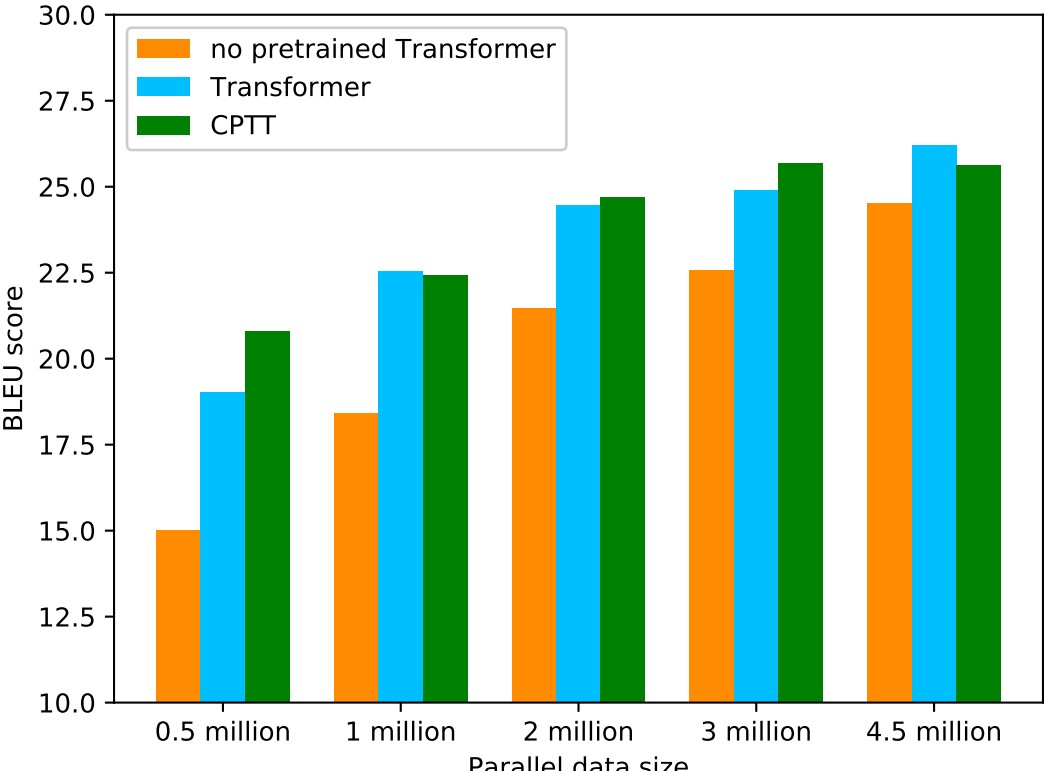

**Figure 6.** Results of different models on parallel corpus of different sizes.

## 6. Conclusions

To keep models consistent between pretraining and translation stages in pretraining fine-tuning mode, we propose using layer-wise coordination transformer and consistent pretraining translation transformer to replace the vanilla transformer as the NMT model. Both models can ensure that all parameters in the NMT model are initialized by the pretrained model. Experimental results on Chinese-to-English and English-to-German translation tasks show our models can obtain better performances with fewer parameters when the number of parallel corpus is small. Through ablation experiments, we found that the source language has a greater impact on the translation results in the pretraining, but cross-language pretraining is more helpful to the translation models. In the future, we plan to explore more pretraining models to utilize monolingual data more effectively for better performance in low resource neural machine translation.

**Author Contributions:** Conceptualization, W.Z.; investigation, G.L., W.Z.; methodology, R.D., X.L.; software, W.Z.; validation, Y.Y., R.D. and W.Z.; funding acquisition, X.L., Y.Y. All authors have read and agreed to the published version of the manuscript.

**Funding:** This work is supported in part by the Xinjiang Uygur Autonomous Region Level talent introduction project (Y839031201), The National Natural Science Foundation of China (U1703133), The Subsidy of the Youth Innovation Promotion Association of the Chinese Academy of Sciences (2017472), the Xinjiang Key Laboratory Fund (Grant No. 2018D04018).

**Acknowledgments:** The authors would like to thank the editor in chief and all anonymous reviewers for their constructive advices.

**Conflicts of Interest:** The authors declare no conflict of interest.

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
