# Peer review of "Keeping Models Consistent between Pretraining and Translation for Low-Resource Neural Machine Translation"

_futureinternet, doi:10.3390/fi12120215_

Round 1
Reviewer 1 Report
The paper is well written and the description of used methodology, experiments and results is clearly presented.
In the paper authors use the mask language modeling as the pretrained task and two transformer variants as the translation model, namely, layer-wise coordination and consistent pretraining translation transformer. These models ensure that all parameters can be initialized by the pretrained model.
The authors have presented the architecture of the used models. The layer in layer-wise coordination Transformer has bee described in details as well as the layer in consistent pretraining Transformer decoder. In the experiments authors simulate the low-resource situation using only a part of the corpus in the translation phase.
The data set and the performed experiments are well presented. The authors used standard metrics in evaluation phase of obtained translation results.
Please correct the paper according to the guidelines:
The source paper for Transformer-based NMT in chapter 3.2. should be referenced.
Language:
Language of the paper has to be improved:
Line 37 The sentence has to begin with capital letter.
Line 57 et al -> et al.
Literature:
Reference list should contain at least several papers presenting other authors recent researches. At the moment the list contains only recent work from 2018.
Technical quality of the paper is satisfactory.
Reviewer 2 Report
The paper focuses on the topic of machine translation between languages when large-scale parallel corpuses are not available.
The topic in interesting and worth investigating. The literature review is relatively brief and could be extended to better contextualize the study. While the approach is interesting, further details would be required in order to make the paper technically sound. Also, the paper uses a variety of concepts, that are not sufficiently explained in order to make the information accessible to readers less familiar to the subject.
In the introduction the notion of language model should be explained. The notion of "masked language modeling" should also be better explained.
The purpose of the "encoder" and "decoder" in the case of a transformer should be explained
At line 29 the paper states that "the decoder of Transformer has more parameters than the encoder". The authors are kindly asked to better explain why this happens.
The authors are kindly asked to define the concept of "mixed-attention". Moreover, all the components of figure 1 should be better explained.
In section 4.2 the paper claims that "When initialized by the same pretrained model, the number of parameters of layer-wise coordination Transformer is only half less than that of Transformer". The reasons for this claim are unclear and should be better explained.
Round 2
Reviewer 2 Report
I would like to thank the authors for thoroughly addressing the comments in the previous review.
Author Response
Thanks
This manuscript is a resubmission of an earlier submission. The following is a list of the peer review reports and author responses from that submission.